

# Ocean warming is the key filter for successful colonization of the migrant octocoral *Melithaea erythraea* (Ehrenberg, 1834) in the Eastern Mediterranean Sea

Michal Grossowicz[1,2,3,*], Or M. Bialik[4,5,*], Eli Shemesh[1], Dan Tchernov[1], Hubert B. Vonhof[6] and Guy Sisma-Ventura[7]

[1] Department of Marine Biology, L.H. Charney School of Marine Sciences, University of Haifa, Haifa, Israel
[2] Yigal Allon Kinneret Limnological Laboratory, Israel Oceanographic and Limnological Research, Haifa, Israel
[3] Biogeochemical Modelling, GEOMAR Helmholtz Centre for Ocean Research Kiel, Kiel, Germany
[4] Department of Marine Geosciences, L.H. Charney School of Marine Sciences, University of Haifa, Haifa, Israel
[5] Institute of Geology, CEN, Universität Hamburg, Hamburg, Germany
[6] Max Plank Institute for Chemistry, Mainz, Germany
[7] National Oceanography Institute, Israel Oceanographic and Limnological Research, Haifa, Israel
[*] These authors contributed equally to this work.

Corresponding author
Michal Grossowicz,
mgrossowicz@geomar.de

## ABSTRACT

Climate, which sets broad limits for migrating species, is considered a key filter to species migration between contrasting marine environments. The Southeast Mediterranean Sea (SEMS) is one of the regions where ocean temperatures are rising the fastest under recent climate change. Also, it is the most vulnerable marine region to species introductions. Here, we explore the factors which enabled the colonization of the endemic Red Sea octocoral *Melithaea erythraea* (Ehrenberg, 1834) along the SEMS coast, using sclerite oxygen and carbon stable isotope composition ($\delta^{18}O_{SC}$ and $\delta^{13}C_{SC}$), morphology, and crystallography. The unique conditions presented by the SEMS include a greater temperature range ($\sim$15 °C) and ultra-oligotrophy, and these are reflected by the lower $\delta^{13}C_{SC}$ values. This is indicative of a larger metabolic carbon intake during calcification, as well as an increase in crystal size, a decrease of octocoral wart density and thickness of the migrating octocoral sclerites compared to the Red Sea samples. This suggests increased stress conditions, affecting sclerite deposition of the SEMS migrating octocoral. The $\delta^{18}O_{sc}$ range of the migrating *M. erythraea* indicates a preference for warm water sclerite deposition, similar to the native depositional temperature range of 21–28 °C. These findings are associated with the observed increase of minimum temperatures in winter for this region, at a rate of $0.35 \pm 0.27$ °C decade$^{-1}$ over the last 30 years, and thus the region is becoming more hospitable to the Indo-Pacific *M. erythraea*. This study shows a clear case study of "tropicalization" of the Mediterranean Sea due to recent warming.

## INTRODUCTION

Increasing global temperatures caused by recent climate change may impose a dramatic effect on the structure and function of ecosystems worldwide (*Lejeusne et al., 2009*; *Burrows et al., 2011*). Long-term records suggest that the greatest impact of climate change on biotic communities might be due to shifts in the maximum and minimum temperatures as well as short-term climatic events, rather than changes in mean annual temperatures (*Stachowicz et al., 2002*). Temperature, as a function of climate, is considered a key filter that could determine the success of introduced marine species (*Theoharides & Dukes, 2007*). Introduced species are defined as living outside their native distributional range through deliberate or accidental human activity. The thermal regime sets broad limits on the distribution of the introduced species that may cause such taxa to fail at the early stages of settlement (*Hewitt & Hayes, 2002*; *Mack et al., 2000*). Under the right environmental conditions and ecosystem fragility, an introduced species may become invasive, i.e., a pest in its new location, which spreads by natural means (*Ehrenfeld, 2010*). Understanding how these long-term fluctuations in environmental conditions facilitate the introduction and successful colonization is of prime importance for developing better predictions regarding the ecological effects of future climate change.

The southeastern Mediterranean Sea (SEMS), which is one of the most rapidly warming regions under recent climate change (*Sisma-Ventura, Yam & Shemesh, 2014*; *Ozer et al., 2017*), offers a natural laboratory to study the process of species introduction in the context of global warming (*Béthoux et al., 1999*). Recent field studies have shown that increased maximum temperatures in the Mediterranean have led, *inter alia*, to multi-species collapse (*Rilov, 2016*), increased seagrass mortality (*Jordà, Marbà & Duarte, 2012*), and a general shift to 'warm-water' species (*Chevaldonné & Lejeusne, 2003*; *Lejeusne et al., 2009*; *Raitsos et al., 2010*; *Rilov & Galil, 2009*). This process was previously defined as ''tropicalization'' of the Mediterranean fauna (*Bianchi & Morri, 2003*).

Increased stratification, due to the recent warming of the eastern Mediterranean surface layer (*Ozer et al., 2017*; *Sisma-Ventura et al., 2017*), as well as damming of its main freshwater sources (*Ludwig et al., 2009*; *Bialik & Sisma-Ventura, 2016*), resulted in a severe nutrient deficiency (*Krom et al., 1991*), leading to an ultra-oligotrophic state (*Azov, 1991*; *Sisma-Ventura, Yam & Shemesh, 2014*; *Hazan et al., 2018*). The response of migrating species to these two simultaneous and rapid processes (warming and elevated oligotrophy) is not well understood.

The introduction of the Indo-Pacific octocoral *Melithaea erythraea* (Ehrenberg, 1834) (Alcyonacea: Melithaeidae) to the SEMS coast was first documented in 1999 within the Hadera power plant harbor (32.47 °N/34.88 °E), where it was found in extremely high abundance, mostly on artificial structures (20–80 colonies per 10 m line transect, (*Fine et al., 2005*). However, in the natural habitat of the Red Sea, this coral is rare, both on natural reefs and artificial structures, and found mostly in shaded habitats on vertical reef walls, and in notches. This behavior is very similar to other azooxanthellate that inhabit these niches uninhabited by zooxanthellate corals (*Fabricius & Alderslade, 2001*). For example, the congeneric octocoral *Melithaea biserialis* (Kükenthal, 1913) is found mostly in shaded

habitats, on vertical reef walls of the Red Sea, and in high density on artificial structures such as the oil jetty of Eilat (*Zeevi-Ben Yosef & Benayahu, 1999*). In 2015, colonies of *M. erythraea* were detected for the first time outside of the Hadera power plant harbor in the rocky Nahsholim Bay at a depth of 3.5 m (32.61 °N/34.91 °E, Fig. 1). Those colonies exhibited a 100% genetic similarity to the *M. erythraea* Red Sea specimen (*Grossowicz et al., 2020*). Further observations revealed a stable population along this coast comprising many colonies, all in shaded locations on either vertical walls or in crevices (*Grossowicz et al., 2020*). *Grossowicz et al. (2020)* reported that *M. erythraea* is not yet invasive, however, its population expansion may yet occur, due to a lag between initial introduction and population explosion (see *Rilov, Benayahu & Gasith, 2004*). They hypothesized that the gradual warming of surface water of the SEMS may have contributed towards the survival of *M. erythraea* during the winter, and enabled this species to expand beyond its distributional range. To date, *M. biserialis* has never been recorded in the Mediterranean.

While reef-building stony corals (Scleractinia) form hard and massive aragonite skeletons, octocorals produce spiny internal polycrystalline high-magnesium calcite skeletal elements that are called sclerites, as well as a central axis (*Cohen & McConnaughey, 2003*; *Taubner et al., 2012*; *Fabricius & Alderslade, 2001*). Sclerites are highly variable in shape, size, and articulation, and differ substantially from one species to another. Therefore, these are an important trait in octocoral taxonomy (*Fabricius & Alderslade, 2001*; *Tentori & van Ofwegen, 2011*). Looking at the skeletal characteristics and isotopic composition may provide insights in coral's ecophysiology (*Chaabane et al., 2016*; *Chaabane et al., 2019*) and therefore may help us to understand the colonization of *M. erythraea* along the SEMS, from a calcification point of view.

The morphological variability of a species' sclerites can be related to its geographical and ecological environment, as was observed in several gorgonians (e.g., *Pseudopterogorgia elisabethae* (*Gutiérrez-Rodríguez et al., 2009*), and *Eunicea flexuosa* (*Prada, Schizas & Yoshioka, 2008*)). Variation in sclerite morphology may be altered in response to depth, water motion, light levels, and environmental factors such as predation pressure (*West, 1997*). Morphological differences may be a response to environmental factors (*Rowley, 2018*), but can also be attributed to accumulated genetic differences, due to the disruption of gene flow among populations (*Prada, Schizas & Yoshioka, 2008*).

The sclerite calcification depends on the physiological traits of each octocoral and the ambient environment (*West, 1997*). For example, a recent study has shown that the red octocoral *Corallium rubrum* calcifcation is not pH upregulated with respect to the ambient seawater, contrary to what is observed in scleractinians (*Le Goff et al., 2017*), making octocoral a highly vulnerable species to enviromental changes, such as a decrease in seawater pH. Growth rates of corals, and octocorals in particular, are positively correlated with temperature (*Chaabane et al., 2019* and references therein). Furthermore, temperature affects the sclerite deposition, as was found in the cold-water octocoral *Primnoa pacifica*, where the magnesium/calcite ratio in the sclerite was positively correlated to water temperature (*Matsumoto, 2007*).

In this study, we test the hypothesis that *M. erythraea* could survive in the Mediterranean Sea due to warming of winter minimum temperatures. This hypothesis will be tested by
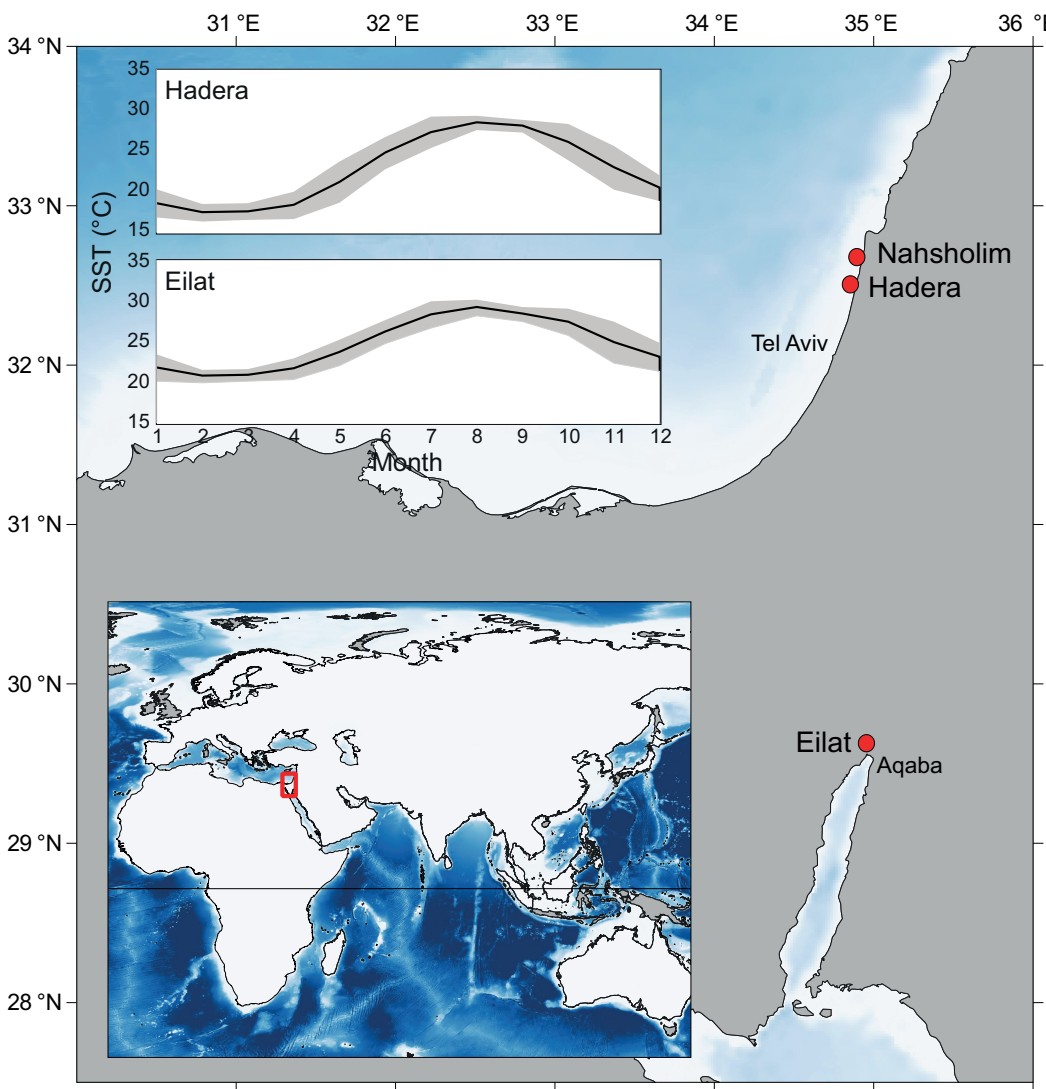

**Figure 1** **Map of the study area and collection sites along the southeastern Mediterranean Sea (SEMS) coast (Hadera and Nahsholim,** *M. erythraea*) **and the Red Sea, Gulf of Aqaba (Eilat,** *M. erythraea* **and** *M. biserialis*). Note the current annual sea surface temperature (SST) range in the SEMS and the northern Red Sea (Tel Aviv and Aqaba, respectively). Maps were created using QGIS (*QGIS Team, 2016*), and ETOPO2 (*National Oceanic and Atmospheric Administration, 2001*) was used as a base layer.

analyzing the sclerites' characteristics using scanned electron microscopy (SEM), carbon and oxygen stable isotopes, and X-ray crystallography to understand, from a calcification perspective, the factors which enabled *M. erythraea* to survive and establish a flourishing population in the SEMS.

## METHODS

### Study site, temperature, coral collection and sclerite isolation

Specimens for SEM examination, crystallography, and carbon and oxygen stable isotope analysis were collected from both the Mediterranean Sea and the Red Sea (Fig. 1). *Melithaea erythraea* branches were collected during scuba dives in June 2016 and May 2017 in Nahsholim Bay. Samples from Hadera port were collected in June 2016, and from Eilat (Red Sea), branches were collected in May 2017. Branches from *M. biserialis* were also collected from Eilat in May 2017. From all collected colonies, a branch from the distal parts of the colony was removed using scissors and preserved in absolute ethanol (96%) prior to examination. For comparison, a specimen of *M. erythraea* from *Fine et al. (2005)*, collected in Hadera port, was examined as well (collected by Y. Aluma in 2002 and stored in formalin at the Steinhardt Museum of Natural History, Israel National Center for Biodiversity Studies, Tel Aviv, Israel).

Branches from each test colony were sub-sampled (1 cm in length). Sclerites were separated from the soft tissue by placing each sub-sample in Eppendorf tubes filled with 10% sodium hypochlorite until the soft tissue was dissolved. After 30 min, the organic debris was removed, and the sclerites were rinsed with distilled water several times to wash off the excess bleach and supernatant. Multiple sclerites were ground to homogenous powders for later isotopic analysis.

Information of ambient conditions (temperature) was collected from Israel Oceanographic and Limnological Research (IOLR) monitoring station in Hadera. For the period of 1994–2004, temperature measurements were taken from a bottom-mounted Paroscientific-8DP060 ADCP. For the period of 2004–2018, temperature measurements were taken from a bottom-mounted 600 kHz WorkHorse Monitor ADCP. The ADCP was located at 11.6 m until 2004, then relocated to 26 m depth, southwest of the easternmost edge of the coal terminals in Hadera. The reported temperature sensor precision is $\pm 0.4\,°C$. The data series was curated for outliers, smoothed and binned.

All samples were collected under the permit from the Israel Nature and Parks Authority (permit number: 2016-18/42200). Conducting the surveys in their areas was approved by Port of Hadera Authority and Eilat-Ashqelon Pipe-Line Company.

### Isotope ratio mass-spectrometry (IRMS)

The fractionation of the oxygen isotopes into biogenic carbonates is a function of ambient temperature and isotopic composition of the seawater at the time of their formation (*Grossman & Ku, 1986*; *McConnaughey, 1989a*; *McConnaughey, 1989b*; *Kim & O'Neil, 1997*). If the isotopic ratio of $^{16}O$ and $^{18}O$ (expressed as $\delta^{18}O$) of both calcium carbonate and water is known, the temperature at deposition can be calculated. The isotopic fractionation of carbon ($\delta^{13}C$) in skeletal material provides information about the organism's metabolism, as well as nutritional information (*McConnaughey et al., 1997*) and, thus, can provide insights into the calcific response of migrating calcifying species to severe oligotrophic conditions.

Stable isotope ($\delta^{18}O$ and $\delta^{13}C$) measurements on bulk powders were performed at the stable isotope laboratory of the Max-Planck Institute for Chemistry, Mainz, on a Thermo

Delta V mass spectrometer interfaced with a Gasbench preparation unit. Sample digestion took place on-line, in >99% orthophosphoric acid, at 70 °C. Coral samples were analyzed together with several calcite standards, including the international standard IAEA603. The reproducibility of these routinely analyzed in-house $CaCO_3$ standards is typically $\leq 0.1‰$ (1 SD) for both carbon and oxygen isotope ratios. Both $\delta^{18}O$ and $\delta^{13}C$ of the sclerites are reported relative to the Vienna Pee Dee Belemnite (VPDB) standard scale.

### Estimation of $CaCO_3$ depositional temperatures

The oxygen isotope composition of biogenic $CaCO_3$ is a function of ambient water temperature and the $\delta^{18}O$ of the seawater at the time of its formation and from this depositional temperatures can be estimated (*Grossman & Ku, 1986*; *Kim & O'Neil, 1997*). The calcite temperature-dependent fractionation during bio-mineralization is described by the equation of *Friedman & O'Neil (1977)*:

$$10^3 \ln \alpha_{calcite-water} = 2.78 \left( 10^6/T^2 \right) - 2.89 \tag{1}$$

where $\alpha_{calcite-water}$ is the oxygen isotope fractionation factor between calcite and water (Eq. (2)) and T is the water temperature (K).

$$\alpha_{calcite-water} = (10^3 + \delta^{18}O_{calcite})/(10^3 + \delta^{18}O_{water}) \tag{2}$$

The mean annual $\delta^{18}O_{sw}$ value of 1.6‰ (*Sisma-Ventura, Yam & Shemesh, 2014*; *Sisma-Ventura et al., 2016*) and 1.9‰ (*Mizrachi et al., 2010*) of surface water in the Mediterranean and the Gulf of Aqaba, respectively, and the $\delta^{18}O_{SC}$ of *M. erythraea* from both habitats were used for the calculation of deposition temperatures. It is noted that the $\delta^{18}O_{sw}$ in both the Mediterranean Sea and the Gulf of Aqaba fluctuated by less than 0.5‰, annually. A combination of the analytical uncertainties of >0.5‰ for both measurements translates into uncertainty of ~2 °C (i.e., 0.2‰ °C $^{-1}$).

### Estimation of percentage of metabolic carbon intake during calcification

We estimated the percentage of the metabolic carbon that contributed to $\delta^{13}C_{SC}$ using the mass balance equation (*McConnaughey et al., 1997*):

$$\delta^{13}C_{calcite} - \varepsilon_{calcite-bicarbonate} = M \left( \delta^{13}C_{Food} \right) + (1-M) \delta^{13}C_{DIC} \tag{3}$$

where M is the percentage of the metabolic carbon contribution and $\varepsilon_{calcite-bicarbonate}$ is the enrichment factor between calcite and bicarbonate (+1‰, *Romanek, Grossman & Morse, 1992*). The $\delta^{13}C$ of *M. erythraea* sclerites may indicate a food source (phytoplankton-small zooplankton (*Zeevi-Ben Yosef & Benayahu, 1999*); $\delta^{13}C \approx -20‰$), and DIC (1‰; *Mizrachi et al., 2010*; *Sisma-Ventura, Yam & Shemesh, 2014*; *Sisma-Ventura et al., 2016*) of both habitats were used to calculate the metabolic contribution to the skeletal buildup.

### X-ray diffraction crystallography

Crystallinity is an important parameter of mineral aggregates such as skeletons. This property can be effected by internal heterogeneity in the crystal, nucleation rate, protein

**Table 1  Summary table of morphometrics, stable isotopic data, and crystallography.** Detailed results can be found in the Supplemental Information.

|  | Length-to-width ratio | *n* | Warts density (# $\mu m^{-1}$) | *n* | FWHM ($n=1$) | $\delta^{13}C$ (‰) | $\delta^{18}O$ (‰) | *n* |
|---|---|---|---|---|---|---|---|---|
| *M. erythraea* |  |  |  |  |  |  |  |  |
| Nahsholim 2017 | $4.97 \pm 1.52$ | 43 | $0.051 \pm 0.007$ | 30 | 0.10 | $-1.159 \pm 0.12$ | $0.363 \pm 0.03$ | 2 |
| Hadera 2016 | $4.14 \pm 1.30$ | 46 | $0.049 \pm 0.006$ | 29 | 0.17 | $-1.242 \pm 0.18$ | $-1.069 \pm 0.04$ | 2 |
| Nahsholim 2016 | $4.77 \pm 1.18$ | 36 | $0.049 \pm 0.006$ | 30 | 0.12 | $-1.380 \pm 0.06$ | $0.207 \pm 0.03$ | 2 |
| Hadera 2002 | $4.55 \pm 1.30$ | 32 | $0.047 \pm 0.006$ | 22 | 0.22 | $-0.381 \pm 0.10$ | $0.331 \pm 0.05$ | 2 |
| Eilat 2017 | $3.18 \pm 1.15$ | 36 | $0.057 \pm 0.013$ | 23 | 0.27 | $-0.250 \pm 0.02$ | $0.033 \pm 0.03$ | 2 |
| *M. biserialis* |  |  |  |  |  |  |  |  |
| Eilat 2017 | $3.58 \pm 2.23$ | 41 | – | – | – | $0.251 \pm 0.03$ | $0.01 \pm 0.04$ | 2 |

framework structure, Sr and Mg concentration and crystal growth rate and is, therefore, a useful parameter to understand environmental effects expressed by in the calcification.

X-ray diffractometry (XRD) was used to evaluate the crystallinity of the sclerites. Full width at high maximum (FWHM) of crystallinity level (*Patterson, 1939*) was calculated for the calcite's d$_{104}$ peak following the Scherrer equation (*Scherrer, 1918*):

$$\tau = \frac{K\lambda}{\beta cos\theta} \tag{4}$$

where $\tau$ is the mean size of the ordered (crystalline) domains, $\lambda$ X-ray wavelength; K is the shape factor; $\beta$ is FWHM and $\theta$ is the Bragg angle. As all parameters are constant other than the FWHM, then $\tau$ is proportional to $\beta^{-1}$. Given that the shape factor could not be determined in most case, FWHM can be used as an index to the level of crystallinity.

The analysis was conducted with a Rigaku MiniFlex benchtop XRD, with the sclerites deposited from suspension on a custom slide and allowed to dry in a desiccator. Diffraction was carried out from 10 to 75° at 0.01° steps at a rate of 2.15° per minute.

## Sclerite morphology and statistical analysis

The morphometric complexity of the sclerites was assessed by SEM analysis. Spindle-shaped sclerites from all specimens were placed in non-coated high/low vacuum mode and were examined and photographed with Jeol JCM-7000 NeoScope benchtop SEM with secondary electron and backscatter modes set to magnify at ×200 and ×300. From each specimen, $n=32$–46 sclerites (see Table 1) were examined, and the measurements of the axes (length and width) were taken using built-in software. In addition, along the longitudinal axis, warts were counted to obtain their density. The number of warts was divided by axis length (in $\mu m$).

Statistical analyses were performed under permutation concepts, non-parametric tests that analyze quantified data that do not satisfy the assumptions underlying traditional parametric tests (e.g., normality, etc., *Collingridge, 2013*). To compare the effect of "site" (Nahsholim, Hadera, and Eilat), and "axis" (long, short) on sclerite morphometrics, we performed a nested Permutation ANOVA, followed by a pairwise permutation posthoc test. Before analysis, data were normalized. Axes' ratio and wart density on sclerites among

sites were examined with one-way Permutation ANOVA, followed by pairwise permutation posthoc test. All values are presented at a confidence interval of 95%. All statistical and multivariate analyses were performed with R i386 3.3.3 (*R Core Team, 2014*) using 'lmPerm' and 'rcompanion' packages.

## RESULTS

### Sclerite morphometrics

The sclerites (shown in Fig. 2) captured in the SEM were mostly spindle shaped, a shape found in all samples. In the Red Sea, we also observed a spheroidal shape (for both *M. erythraea* and *M. biserialis*). The sclerites exhibited some visual differences between the sites. The Red Sea sclerites looked thicker than the Mediterranean ones, and their wart density appear to be higher (see Table 1, raw measurement of the sclerites is provided in the Supplemental Information).

The sclerite axis characteristics were significantly different by size (Fig. 3A, Nested permutation ANOVA, $p < 0.001$), where posthoc pairwise comparisons suggested that the corals in the Red Sea have thicker sclerites in comparison with all other test corals collected in the Mediterranean (all pairings with the Eilat specimen, except Eilat-Hadera 2002, are $p$-adjusted $< 0.002$).

These differences are also reflected in the long-to-short axes ratio comparison (Fig. 3B, Permutation ANOVA, $p < 0.001$), where the Red Sea sclerites had the smallest ratio ($3.19 \pm 1.15$) compared to all other sites ($4.78 \pm 1.18$, and $4.98 \pm 1.52$ in Nahsholim 2016 and 2017, respectively, and $4.55 \pm 1.30$, $4.14 \pm 1.30$ in Hadera 2002 and 2016, $p$-adjusted $< 0.005$), not including the *M. biserialis* pair ($p$-adjusted $= 0.39$). *M. biserialis*' ratio ($3.58 \pm 2.23$) was similar to both Hadera (2002 and 2016) specimens ($p$-adjusted $> 0.2$) and was significantly different than all the other sites ($p$-adjusted $< 0.02$). In addition, the Hadera 2016 sclerite axis ratio is significantly different than those collected from Nahsholim ($p$-adjusted $< 0.05$).

Warts along the sclerites' axis were counted to determine their density. The wart density differed among the different sites (Fig. 3C, Permutation ANOVA, $p < 0.001$), where the *M. erythraea* Red Sea specimen has a significantly higher density ($0.057 \pm 0.013 \ \# \ \mu m^{-1}$) than those collected from the Mediterranean ($0.049 \pm 0.006$, $0.051 \pm 0.007$, $0.0047 \pm 0.008$ in Nahsholim 2016–2017, and $0.047 \pm 0.006$ and $0.049 \pm 0.006 \ \# \ \mu m^{-1}$ in Hadera 2002 and 2016, respectively, $p$-adjusted $< 0.05$), except the pair collected from Eilat and Nahsholim 2017 ($p$-adjusted $> 0.1$).

### Isotopes

The results in Fig. 4 summarize the $\delta^{18}O_{SC}$ and $\delta^{13}C$ values of *M. erythraea* and *M. biserialis* sclerites from the Gulf of Aqaba (Red Sea), and of *M. erythraea* from the Israeli coast (SEMS). The $\delta^{18}O_{SC}$ value of samples from the Gulf of Aqaba, which were collected during May 2017, ranged between $0.33 \pm 0.045‰$ in *M. erythraea* and $0.01 \pm 0.061‰$ in *M. biserialis* (the average temperature of deposition was 23 °C and $24.5 \pm 0.5$ °C, respectively). Similarly, *M. erythraea* samples from the Port of Hadera (the first documented colonization), was collected in the spring of 2002, and yielded $\delta^{18}O_{SC}$ values between 0.26

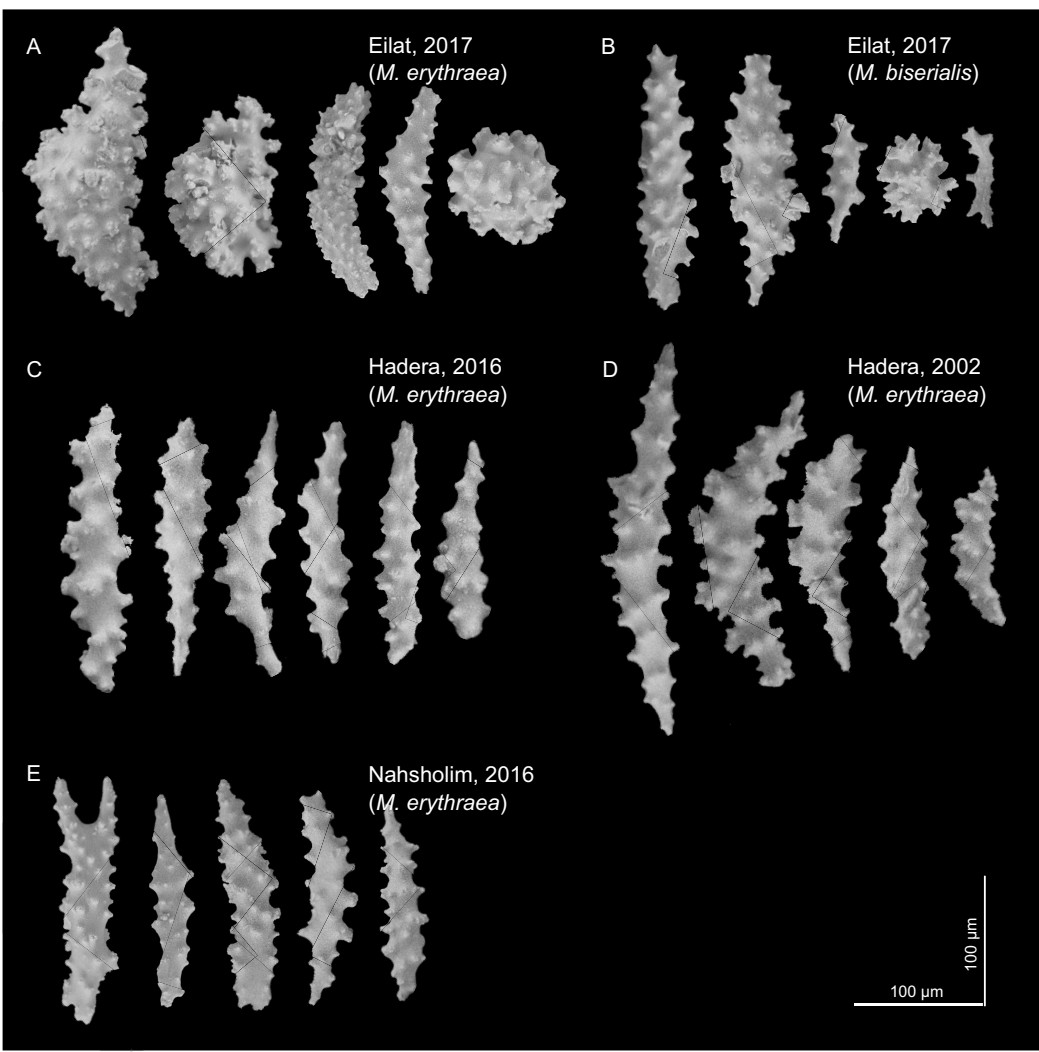

**Figure 2** *M. erythraea.* and *M. biserialis* **sclerites morphology.** (A) Spindles and spheroid of *M. erythraea* from Eilat (2017); (B) spindles and spheroid of *M. biserialis* from Eilat (2017); (C) spindles of *M. erythraea* from Hadera (2016); (D) spindles of *M. erythraea* from Hadera (2002); and (E) spindles of *M. erythraea* from Nahsholim (2016). Please note the thicker and denser warts of the Red Sea sclerites with respect to the slender Mediterranean counterparts. Information on the coral sclerites can be found in *Kükenthal (1913)*.

and $0.4 \pm 0.063‰$ (depositional temperatures of $21 \pm 0.7\,°C$). The $\delta^{18}O_{SC}$ values of samples collected during 2016 and 2017 in early spring from Hadera and Nahsholim Bay ranged between $0.22 \pm 0.03$ and $0.36 \pm 0.052‰$ (deposition temperatures of $21.2 \pm 0.5\,°C$). The 2017 (early summer) samples from the Nahsholim Bay yielded $\delta^{18}O_{SC}$ values of $1.07 \pm 0.05‰$ (deposition temperatures of $27.7 \pm 0.4\,°C$).

The samples from the Gulf of Aqaba (May 2017) yielded $\delta^{13}C_{SC}$ values ranging between $0.25 \pm 0.033‰$ (*M. biserialis*) and $-0.26 \pm 0.02‰$ (*M. erythraea*), and those sampled from the Hadera port in 2002 averaged $-0.38 \pm 0.1‰$. The $\delta^{13}C_{SC}$ values of samples

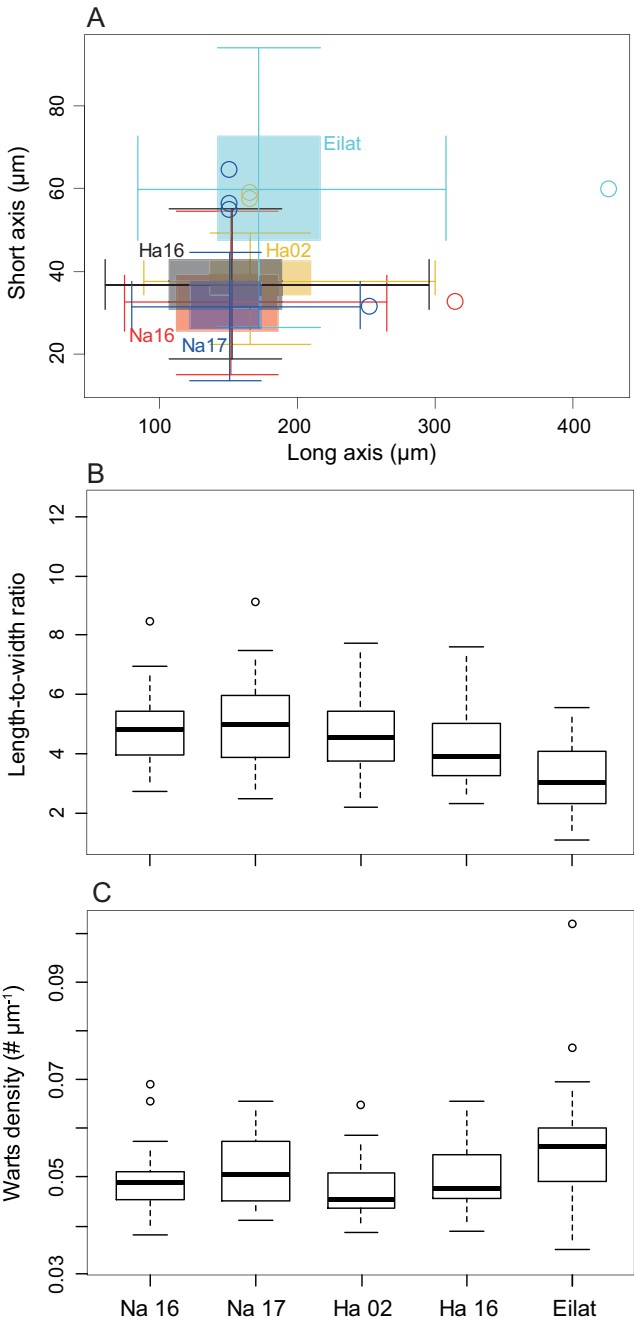

**Figure 3 Sclerite morphometrics (*M. erythraea*).** (A) Comparison of sclerite width and length from the different sites; (B) Sclerites length-to-width ratio at the different sites; and (C) Wart density along the main sclerites axes at the different sites. Na –Nahsholim, Ha –Hadera.

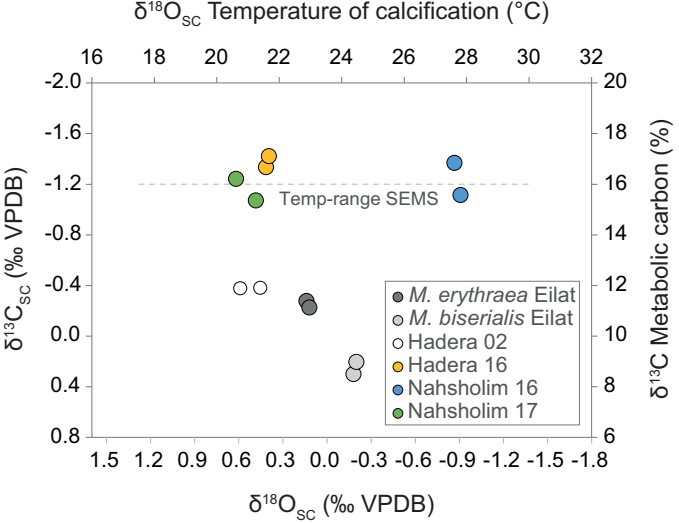

**Figure 4** **Cross plot of $\delta^{13}$C derived metabolic carbon (Eq. (3)) and $\delta^{18}$O derived temperature (Eq. (1)) of sclerite deposition of *Melithaea erythraea*.** The mean offset of $-1.26‰$ in $\delta^{13}C_{SC}$ between the Red Sea samples ($\delta^{13}$C values between $-0.28$ and $0.30‰$) and the samples collected from the southeastern Mediterranean Sea (SEMS) coast in 2016 and 2017 ($\delta^{13}$C values between $-1.07$ and $-1.42‰$), represent an increase of $\sim$50% in metabolic carbon intake during calcification. The 2002 $\delta^{13}C_{SC}$ value of $-0.38‰$ from Hadera (SEMS) shows an intermediate value between the recent SEMS and Red Sea specimens.

collected during 2016 and 2017 in early spring from Hadera and Nahsholim Bay ranged between $-1.38 \pm 0.062$ and $-1.16 \pm 0.12‰$. The 2017 early summer samples from the Nahsholim Bay yielded $\delta^{13}C_{SC}$ values, averaging $-1.24 \pm 0.18‰$. Results are summarized in Table 1 and the 'Isotopes' section within the Supplemental Information.

## Crystallography

Full width at high maximum (FWMH) of the $d_{104}$ peak of the calcites ranged from 0.10 to 0.27 with the d spacing ranging from 2.988 to 2.997; peak asymmetry ranged from 0.56 to 2.8. The Eilat samples and 2002 Hadera samples exhibit the lower values of asymmetry and d spacing values with the higher FWHM values relative to the 2016 and 2017 values. The length of $\delta^{13}C_{SC}$ and the long axis are positively correlated to FWMH (inversed to crystallinity, Table 1, Fig. 5).

## Sea surface temperature (SST)

Mean SST in the coast of Hadera had not changed significantly since the early 1990's and remained at $23.2 \pm 4.3$ °C ($n = 199229$). However, this figure is misleading as the extreme temperatures have shifted in both summer and winter (Fig. 6). Notably, minima temperature (10th percentile) has increased by $\sim$2 °C while maxima temperature (90th percentile) has diminished by $\sim$1 °C. In addition, the fractional time during which temperature was below 18 °C (1994's 20th percentile) has diminished from $\sim$18% in 1994 to less than 5% in 2019, while the fraction above 28 °C (1994's 80th percentile) has diminished from $\sim$24% in 1994 to $\sim$15% in 2019.

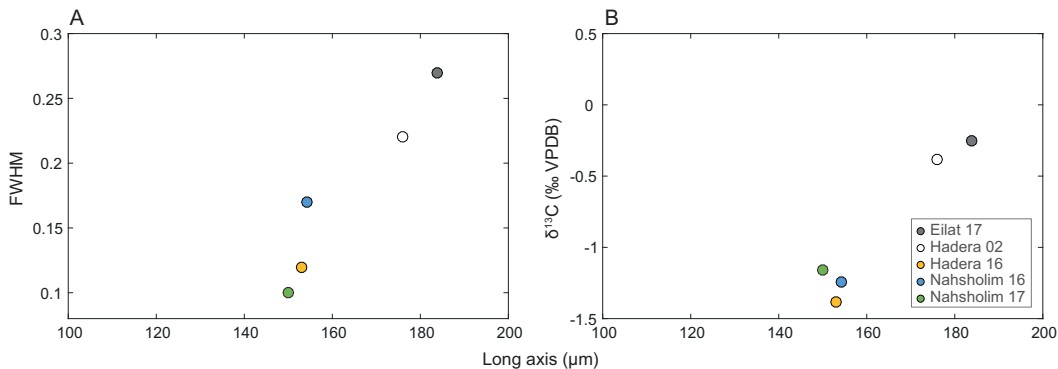

**Figure 5** **Relation of the *M. erythraea* sclerite morphometry (length of the long axis).** to (A) its full width at half maximum (FWHM) of the calcite [104] peak, and (B) $\delta^{13}C$ values of the bulkscleritesfor each year.

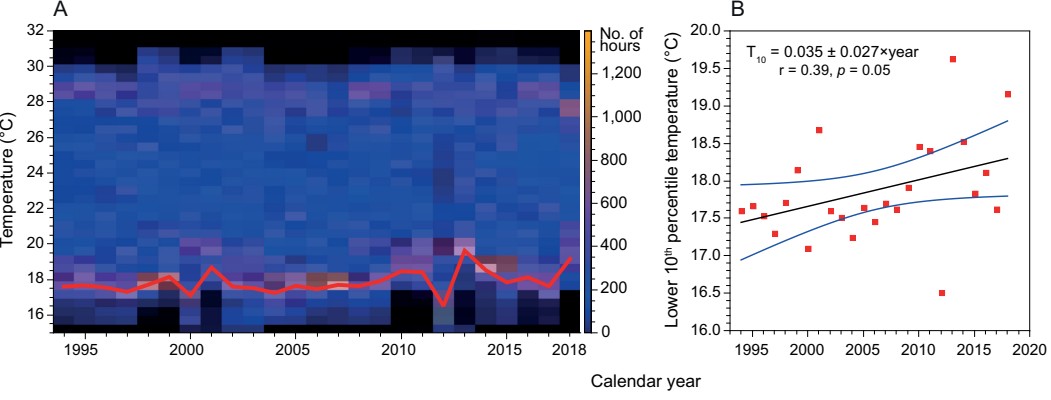

**Figure 6** **Variation in sea surface temperature of the Israeli coast (Hadera monitoring station time series, 1994 to 2018).** (A) Stacked annual histograms, the red line traces the lower 10th percentile; (B) Fitted trendline of recorded temperature illustrates the increased warming of water during winters (annually, for the lower 10th percentile).

# DISCUSSION

Within the last few decades, the effects of anthropogenic global warming have become more pronounced, to the point they can no longer be ignored or attributed to natural processes (*Weart, 2009*). One of these effects is the opening of previously unavailable geographical niches to invasive species (*Bianchi & Morri, 2003*). This is expressed most intensively in areas experiencing extreme warming, such as higher latitudes (*Stachowicz et al., 2002*), although similar processes also occur in some lower latitude domains such as the SEMS (*Rilov & Galil, 2009*). Overall, the Levantine Sea has experienced warming of between ~0.02 (*Sisma-Ventura, Yam & Shemesh, 2014*; *Marbà et al., 2015*) to ~1.0 °C yr$^{-1}$ (*Ozer et al., 2017*), with the highest rates of warming in the last few decades. This shift

has also modified the coastal water's minimum winter temperatures, which have risen from ∼16 to ∼18 °C since the 1990s (*Ozer et al., 2017*).

As was observed in several gorgonian species, the variation in morphological features is also related to the geographical environment (*Gutiérrez-Rodríguez et al., 2009*; *Prada, Schizas & Yoshioka, 2008*). Octocoral sclerite deposition is dependent on the octocoral's physiological traits, and environmental parameters such as temperature (*West, 1997*). The growth rate and deposition of the skeletal elements of octocorals and other calcifiers, such as coralline algae, is correlated with temperature (*Crabbe, 2008*; *Matsumoto, 2007*; *Chaabane et al., 2019*; *Vielzeuf et al., 2018*), and therefore, we believe that upon migration, *M. erythraea* was subject to environmentally-related physiological changes to their sclerite deposition. Sclerite morphology shows that the *M. erythraea* Red Sea specimens are thicker (lower long-to-short axes ratio) with increased wart density in comparison with the Mediterranean conspecific. Furthermore, the specimen from Hadera collected in 2002 may be in a transition state between the recent Mediterranean Sea collections and those from the Red Sea. However, the port of Hadera is a confined body of water influenced by its power plant water discharge and is by no means representative of the natural environment of *M. erythraea*.

The $\delta^{18}O_{SC}$ values of *M. erythraea* collected along the SEMS coast during late winter and early spring and summer yielded depositional temperatures between 21 and 28 °C, respectively. These temperatures match the depositional temperature range of its original habitat in the Red Sea (*Al-Rousan et al., 2007*; *Mizrachi et al., 2010*) and those measured in the native specimens. Thus, the hypothesis that the recent warming of the SEMS by ∼1.0 °C decade$^{-1}$ over the last 30 years (*Ozer et al., 2017*) has enabled the octocoral *M. erythraea* to successfully colonize the area is supported. Despite not having any *M. erythraea* samples during the maximum winter temperatures, our bulk $\delta^{18}O_{SC}$ values, integrating multiple sclerites, indicate a preferential warm water calcification of *M. erythraea*, and further support that the warming of the SEMS surface water is a key factor for its successful migration. This assumption is further supported by the preferential warm water calcification of endemic SEMS species, such as the reef building gastropods *Dendropoma petraeum* complex (*Dendropoma* spp.) and *Vermetus triquetrus* (*Sisma-Ventura, Yam & Shemesh, 2014*).

Stable isotope analysis of *C. rubrum* showed that both $\delta^{18}O$ and $\delta^{13}C$ are strongly influenced by kinetic vital effects, which impede the direct extraction of temperature time-series reconstructions for cold water octocoral (*Chaabane et al., 2016*). However, the results of the study by *Chaabane et al. (2016)* also show that at higher temperatures, octocoral's calcification approaches temperature-dependent equilibrium fractionation, and is less likely affected by vital effects, as is the case for other Mediterranean warm water calcifiers (*Sisma-Ventura, Yam & Shemesh, 2014*). Moreover, while high intra- and inter-annual variations of Mg/Ca were observed in the high-resolution profiles of sclerites of the Mediterranean *C. rubrum*, the mean Mg/Ca composition enabled good estimates of palaeoseawater temperature (*Chaabane et al., 2019*). Similarly, our approach of using bulk powders, integrating multiple sclerites may have reduced the isotopic shifts resulting from the vital effects.
Further, by analyzing time series SST data measured at the monitoring station in Hadera port, we show that the lower 10th percentiles of temperatures (winter minimum temperatures) have increased by almost 2 °C (Fig. 6) over the last three decades. Thus, the warming of the winter minimum temperatures of the SEMS is gradually expanding the time-frame of favorable conditions for the introduction of tropical species.

While recent warming has provided the threshold conditions for the long-term colonization of *M. erythraea*, it is also accompanied by an increase in oligotrophic conditions (*Azov, 1991*; *Sisma-Ventura, Yam & Shemesh, 2014*; *Ozer et al., 2017*). However, it is difficult to capture the effect of the changes in trophic level on the isotopic ratio of the migrating *M. erythraea*. We assume that this factor may impose yet another limitation/stress to the migrating octocoral, which is not likely to be distinguished by the $\delta^{13}C_{SC}$ values, since this limitation may be only secondary to the thermal effect that can trigger periods of rapid growth, resulting in larger metabolic intake during calcification (*McConnaughey, 1989a*; *McConnaughey, 1989b*; *Klein, Lohmann & Thayer, 1996*; *Lorrain et al., 2004*). Nevertheless, the most pronounced modification to the isotopic signal between the present SEMS population and the Red Sea population was observed in the $\delta^{13}C_{SC}$ values. We recorded a mean offset of $-1.26\permil$ in $\delta^{13}C_{SC}$, between the *M. erythraea* Red Sea samples ($\delta^{13}C$ values between $-0.28$ and $0.30\permil$, representing 8.5% of metabolic carbon) and the samples collected from the SEMS coast in 2016 and 2017 ($\delta^{13}C$ values between $-1.07$ and $-1.42\permil$, representing 15.4 to 17.1% of metabolic carbon). Interestingly, the 2002 $\delta^{13}C_{SC}$ value of $-0.38\permil$ (representing 11.9% of metabolic carbon) from Hadera (SEMS) does not show this level of modification but rather showed an intermediate value between the recent SEMS and *M. erythraea* Red Sea specimens.

This modification toward isotopic disequilibrium is common in biological carbonates and is best described by two possible effects: the first is a kinetic effect and the second is a metabolic effect. A kinetic isotope effect, which modulates both the carbon and oxygen isotopic composition simultaneously (*McConnaughey, 1989a*; *Maier, Pätzold & Bak, 2003*), is a known factor that influences the $\delta^{18}O_{SC}$ and $\delta^{13}C_{SC}$ values of cold water octocoral, such as the Mediterranean *C. rubrum* (*Chaabane et al., 2016*). Here, kinetic isotopic effects seem less likely to have been affecting the *M. erythraea* bulk skeletal isotopic composition. This is based on the samples of 2016, that show a wide range of $\delta^{18}O_{SC}$ values, between 0.21 (Hadera) and $-1.07\permil$ (Nahsholim), while the $\delta^{13}C_{SC}$ values were merely unchanged ($-1.31 \pm 0.14\permil$, Hadera and Nahsholim). A metabolic isotope effect, which modulates only the carbon isotopic composition (*McConnaughey, 1989a*), could thus explain the lower $\delta^{13}C$ values of the *M. erythraea* in the SEMS compared to the specimens in their native environment. Moreover, the metabolic isotope effect of *M. erythraea* could be related to respiration, which leads to the incorporation of isotopically-depleted metabolic carbon during sclerite deposition (*McConnaughey et al., 1997*). This is because respiration enriches the internal DIC pool from which the skeleton is precipitated with $^{12}C$.

Other factors influencing the $\delta^{13}C_{SC}$-like changes in diet or the ambient $\delta^{13}C_{DIC}$ range between the two habitats can be ruled out. The soft tissues $\delta^{13}C$ and $\delta^{15}N$ values of both native and introduced species (*Grossowicz et al., 2020*) are similar, indicating no significant change in diet. Furthermore, the ambient $\delta^{13}C_{DIC}$ range in both habitats is very similar

(*Mizrachi et al., 2010*; *Sisma-Ventura, Yam & Shemesh, 2014*; *Sisma-Ventura et al., 2016*). Thus, the change in $\delta^{13}C_{SC}$ of the sclerites should be in the fraction of metabolic carbon incorporate during calcification. The observed increase in metabolic carbon fraction may result from rapid skeletal growth (*McConnaughey, 1989a*; *McConnaughey, 1989b*; *Klein, Lohmann & Thayer, 1996*; *Lorrain et al., 2004*; *Chaabane et al., 2016*; *Chaabane et al., 2019*), or through ontogenetic effects (*McConnaughey & Gillikin, 2008*), which are both known to occur in warm water. Exposure to more stressful conditions may also explain the observed changes, as was suggested for the Mediterranean cold-water octocoral (*Vielzeuf et al., 2018*).

Interestingly, the modification in the $\delta^{13}C_{SC}$ isassociated with crystals becoming larger in the 2016–2018 population relative to that of 2002 and those in Eilat. Crystal size suggests that less carbonate is precipitated in the SEMS specimens. This is also reflected by the higher wart density and axes sizes of the Red Sea samples, compared to those from the Mediterranean ($p$-adjusted $< 0.05$). Keeping in mind that the SEMS is an ultra-oligotrophic region, these changes in the sclerites might be due to calcification under more stressful conditions, resulting from a preference for warmer water and rapid skeletal growth, and increased incorporation of metabolic carbon, as was found in the reef-building gastropods *D. petraeum* complex (*Sisma-Ventura, Yam & Shemesh, 2014*).

## CONCLUSIONS

Our results suggest that the increase of minimum winter temperatures, which is a regional manifestation of global climate change (*Sisma-Ventura, Yam & Shemesh, 2014*; *Amitai et al., 2020*), enabled the successful colonization and recent spreading of tropical *M. erythraea* populations along the SEMS coast, by prolonging the thermally favorable calcification season. This study shows, for the first time, the response of the introduced soft coral *M. erythraea* to the SEMS, a fast-warming and ultra-oligotrophic environment. This observation is part of the overall story of "tropicalization" of the Mediterranean, and it provides insight into how species migrate and colonize under the combined effects of warming surface oceans and increased oligotrophy, driven by global climate change.

## ACKNOWLEDGEMENTS

We would like to thank Dr Yaniv Aluma from Tel Aviv University and Prof. Maoz Fine from the Interuniversity Institute for Marine Sciences in Eilat, Israel, for advising on coral location and presence in the port; Tal Ozer from IOLR for access to raw temperature data; Dr Revital Ben David-Zaslow from the Steinhardt Museum of Natural History, Israel National Center for Biodiversity Studies, Tel Aviv, Israel, for *M. erythraea* specimens; Prof. Yehuda Benayahu from Tel Aviv University and Dr Tali Mass from University of Haifa for fruitful discussion; Eitan Maharam (Northern Wind Diving Center, Nahsholim, Israel), Dr Ateret Shabtay (Technion, Israel), Hagai Nativ, and Ziv Zemah-Shamir (University of Haifa, Israel) for diving support; and Leigh Livne for English editing.

### Funding

Financial support was provided to Michal Grossowicz by the Mediterranean Sea Research Center of Israel (MERCI) consortium. The funders had no role in study design, data collection and analysis, decision to publish, or preparation of the manuscript.

### Grant Disclosures

The following grant information was disclosed by the authors:
Mediterranean Sea Research Center of Israel (MERCI) consortium.

### Competing Interests

The authors declare there are no competing interests.

### Author Contributions

- Michal Grossowicz conceived and designed the experiments, performed the experiments, analyzed the data, prepared figures and/or tables, authored or reviewed drafts of the paper, and approved the final draft.
- Or M. Bialik conceived and designed the experiments, performed the experiments, analyzed the data, prepared figures and/or tables, authored or reviewed drafts of the paper, and approved the final draft.
- Eli Shemesh and Dan Tchernov performed the experiments, authored or reviewed drafts of the paper, field data collection, and approved the final draft.
- Hubert B. Vonhof performed the experiments, authored or reviewed drafts of the paper, isotopic analysis, and approved the final draft.
- Guy Sisma-Ventura conceived and designed the experiments, performed the experiments, analyzed the data, prepared figures and/or tables, authored or reviewed drafts of the paper, and approved the final draft.

### Field Study Permissions

The following information was supplied relating to field study approvals (i.e., approving body and any reference numbers):

Field collection and survey permissions in their areas were given by the Port of Hadera Authority and Eilat-Ashqelon Pipe-Line Company; Permits were given by the Israel Nature and Parks Authority (2016-18/42200).

### Data Availability

The raw measurements and R code are available in the Supplementary Files.

### Supplemental Information

Supplemental information for this article can be found online at http://dx.doi.org/10.7717/peerj.9355#supplemental-information.

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
