# Peer review of "Ocean warming is the key filter for successful colonization of the migrant octocoral Melithaea erythraea (Ehrenberg, 1834) in the Eastern Mediterranean Sea"

_PeerJ, doi:10.7717/peerj.9355_

## Round 0.1 · original submission · Major Revisions

Dear Michal and co-authors,

I have received assessments from three independent reviewers. While all reviewers found the study interesting and worth publishing, two reviewers require some major revisions. In particular, the revised version should be restructured and thoroughly proof-read by a native English speaker (you may decide to invite an additional co-author for this), and care need to be taken on data interpretation with regards to consideration of other published geochemical work on octocorals.

Overall, the reviewers have provided you with excellent suggestions on how to improve the manuscript (please note that Reviewer 2 provided an annotated document with specific comments for you to address), and I will be looking forward to receiving the revised version along with a point-by-point response to each of the reviewers' comments.

With warm regards,
Xavier

Reviewer 1 ·

Basic reporting

THe language is correct and clear
The references are appropriately cited
The manuscript is well written
The question is clearly formulated
The conclusions are congruent with the results

Experimental design

The applied methods are well described
The described research is rigorous
Results are interesting

Validity of the findings

The story is new and interesting
as already reported conclusions are congruent

Additional comments

This is an interesting paper that can be published as it is
I have only a criticism. If spicules of the mediterranean species are different from those of the red sea ones, while authors do not take in consideration the hypothesis that the mediterranean specimens belong to a different species? I think that this idea have to be at least discussed

Reviewer 2 ·

Basic reporting

Manuscript Number: PeerJ 45922

Title: Warming is the key filter for successful colonization of the migrant octocoral Melithaea erythraea in the Eastern Mediterranean

Article Summary:
This manuscript seeks to test the hypothesis that the increase in water temperature in the Eastern Mediterranean, and thus, global climate change, has determined the colonisation success of the Indo-Pacific octocoral Melithaea erythraea (Ehrenberg, 1834). The authors have used stable isotope analysis of oxygen and carbon of the sclerites of this and a comparative species, Melithaea biserialis (Kükenthal, 1908). They have also compared their results with water temperature, although these measurements were taken at depths significantly deeper than the specimens collected for analyses.

Article Review:
Overall, this is a very interesting study and highlights some key results with regard to the subsequent success of migrating species as a consequence of global climate change. This study is worthy of publication, and I enjoyed reading it, thank you. There are key points that need to be addressed prior to acceptance for publication.

Key Points to Review:
1. Foremost, this manuscript requires the thorough review of a native English speaker who regularly writes and reviews publishable text in the English language. There are many confusing sentences and grammatical errors throughout the text, which reduce the quality of the message in the paper.

2. It would benefit the manuscript to define the terms 'introduced', 'invasive', ‘migrant’ and then how they relate to this study. At present, the terms used suggest that the octocoral Melithaea erythraea is detrimental to the marine ecosystems of the SEMS, however, there appears to be no published or presented data to support this.

3. Both the abstract and introduction would benefit from being re-written and structured so that the reader can be clear on the message that the authors are trying to portray, as well as providing the sufficient back ground on the information such as introduced vs. invasive. The hypothesis and objectives were not laid out well. Suggestions have been made throughout the text in the attached pdf.

Experimental design

Research was good.

Research question is not well defined. I think that this is due to English not being the native language of the author(s) and consulting someone who is would help re-define the manuscript significantly.

I don't recall seeing any sample replicate numbers. The number of test samples needs to be specifically stated in the text.

Methods fairly well described.

Validity of the findings

The findings would be clearer and appear more meaningful with the assistance of a native English speaker.

Additional comments

Manuscript Number: PeerJ 45922

Title: Warming is the key filter for successful colonization of the migrant octocoral Melithaea erythraea in the Eastern Mediterranean

Article Summary:
This manuscript seeks to test the hypothesis that the increase in water temperature in the Eastern Mediterranean, and thus, global climate change, has determined the colonisation success of the Indo-Pacific octocoral Melithaea erythraea (Ehrenberg, 1834). The authors have used stable isotope analysis of oxygen and carbon of the sclerites of this and a comparative species, Melithaea biserialis (Kükenthal, 1908). They have also compared their results with water temperature, although these measurements were taken at depths significantly deeper than the specimens collected for analyses.

Article Review:
Overall, this is a very interesting study and highlights some key results with regard to the subsequent success of migrating species as a consequence of global climate change. This study is worthy of publication, and I enjoyed reading it, thank you. There are key points that need to be addressed prior to acceptance for publication.

Key Points to Review:
1. Foremost, this manuscript requires the thorough review of a native English speaker who regularly writes and reviews publishable text in the English language. There are many confusing sentences and grammatical errors throughout the text, which reduce the quality of the message in the paper.

2. It would benefit the manuscript to define the terms 'introduced', 'invasive', ‘migrant’ and then how they relate to this study. At present, the terms used suggest that the octocoral Melithaea erythraea is detrimental to the marine ecosystems of the SEMS, however, there appears to be no published or presented data to support this.

3. Both the abstract and introduction would benefit from being re-written and structured so that the reader can be clear on the message that the authors are trying to portray, as well as providing the sufficient back ground on the information such as introduced vs. invasive. The hypothesis and objectives were not laid out well. Suggestions have been made throughout the text in the attached pdf.

Annotated reviews are not available for download in order to protect the identity of reviewers who chose to remain anonymous.

·

Basic reporting

Warming is the key filter for successful colonization of the migrant octocoral Melithaea erythraea in the Eastern Mediterranean

Basic reporting
Grossowicz et al. investigate the octocoral Melithaea erythraea, endemic in the Red Sea and since 2002 documented as newly introduced species in the south-eastern Mediterranean Sea (SEMS). The authors argue that rising winter water temperature in the SEMS facilitated successful establishment of colonies and that changing trophic levels may play a further role to their advantage. Sclerite morphology and mineralogical composition (XRD) are combined with bulk oxygen isotopes (d18O) and stable carbon isotopes (d13C). The latter are used to reconstruct temperature at the time of calcification from d18O and the degree of metabolic carbon uptake from d13C.

The language is generally clear, but shows a minor albeit systematic issue with use of plural, that needs a second look throughout the document (e.g. “sclerites morphometrics, sclerite morphometrics, sclerite’s morphometrics”). Context of introduction and background are well tailored and simple to follow. Structure and standards of PeerJ are met. But the manuscript would benefit from major revisions prior to publication, including a more in-depth discussion on the validity of the conclusions drawn from the geochemical and mineralogical data. All provided Figures are relevant and necessary, ameliorations to content and captions are given below. References are complete, but would need a format polishing, see detailed line by line comments below. The raw data are all supplied. In the supplementary files the units are missing in the column headers of most parameters (percent, permille, µm, °C, etc.). These data columns need some more description.

Experimental design

Experimental design
The current work provides original data that are worthwhile to be published with PeerJ. Methods are described with sufficient detail for replication and research questions are clear. But data interpretation lacks rigor and relation to other geochemical work on octocorals. In particular the aspect on reconstructed temperature at time of calcification needs a careful look, with respect to sclerite deposition (season, precipitation velocity, ontogeny, vital effects). At the current stage the argumentation remains somewhat circular, with reconstructed temperatures accepted as correct and result in the claim that corals can hence prevail in the basin.

Validity of the findings

Validity of the findings
All underlying data are provided, statistics look sound, and data provide an important mosaic stone for octocoral calcification research. Their impact and novelty with respect to other research still needs to be fathomed, and some references (e.g. Taubner et al., 2012; Vielzeuf et al., 2018) are suggested to build a more in-depth discussion.

Additional comments

General comments
Introduction
Sclerite morphometric data, bulk stable isotopes and mineralogical characteristics are employed to solve how well the newly invaded species can cope with the conditions in the SEMS, with temperature and trophic state as the two key variables. While few would argue that rising winter temperatures would make conditions more suitable, this aspect is not fully followed through. Are the conditions with 21 to 28°C in the Gulf of Aqaba at Eilat representative for the temperature range this species can live in? If there are M. erythraea sites prevailing at colder conditions in the Red Sea, eventually at deeper settings, this would then hence be less surprising for their colonization success?
What about the second variable - trophic levels? It is mentioned that the trophic conditions are changing in the SEMS. It would be easier to follow to compare first the natural trophic level between the Red Sea and SEMS first and then move to temporal changes. Is Eilat more eutrophic?

Material and methods
In the Material and Methods it is not always clear, when you talk about M. biserialis or M. erythraea. For instance, in Lines 127 to 131 – I’d presume this specimen is from Hadera 2002 and hence M. erythraea?
It would be good to add some more information on the sampled colonies. How high were they? Have the distal branches been sampled or the basal stem? Sclerites may form fast, but could continue to add layers for longer time periods. This is an important aspect for the discussion of reconstructed “calcification temperatures” and vital effects. Was there a conspicuous difference in colony morphology between SEMS and the Eilat specimen?
Lines 137 ff, the ADCP unit captures the Hadera port, the Nasholim Bay an Eilat/Aqaba SSTs are derived from a web-source this should be mentioned either here, or put a link to the supporting online material and direct to Fig. 1.
Section 2.2 on Sclerite morphology can be combined with Section 2.7 on Statistical analyzes, since all tests refer to morphometric measurements.
Line 159 this aspect on how d13C are expected to reflect responses to oligotrophic condition needs some more explanation – either here or in the Introduction and Discussion parts. Temperature and trophic level are the key aspect of the paper. While the temperature part is easy to follow, this second aspect is relatively underrepresented in the argumentation.
Section 2.3 mention that for all sites two stable isotope samples have been measured.
Section 2.4 seawater values are given against VSMOW? Needs to be added behind both values (lines 178/179). If seawater values remain within 0.5 ‰ variation, then the temperature uncertainty range (lines 183/184) would be 2°C instead of 0.5°C?
Section 2.6 XRD, you might want to mention the calculation of the %Mg here (for now these values are only in the online material but would be important for the discussion). You could use Titschack et al. (2011) if it was calculated via Rietveld methods, or another reference for 104-calcite peak calculation.

Results
Section 3.1 It might be better to give an overview of the raw measurements first, before moving to “significance of differences” – this would make this paragraph easier to read. It is not immediately clear that M. biserialis has a complete overlap with the field of length/width of M. erythraea and is hence not shown in Figure 3A.
Section 3.2 (Line 246f.) why are both species presented with pooled isotope range? They may have different strengths of the metabolic effect.
M. biserialis misses XRD – can these measurements still be done? This could be interesting because the d13C does not fall onto the same regression as axis length vs d13C as in M. erythraea (Fig. 5). Why is that? According to the calculation its metabolic carbon uptake in percent would be similar to M. erythraea, despite strongly differing sizes.
It would be good to present the calculated Mg-content based on FWHM or d, as well as to point out its correlation with d13C. This could be an indication for a vital effect, since Mg/Ca and d13C are correlated in bamboo corals and also in scleraxonian corals like Corallium rubrum.

Discussion
Sclerite morphometrics need to be discussed also with respect to ontogenetic growth and timing of sclerite deposition. What is the normal known size variability of sclerites in the Red Sea? If there are other taxonomic/morphometric studies from the Red Sea – is the single specimen of M. erythraea from Eilat representative? Is there an ontogenetic effect on sclerite morphology? It would be good to relate this to colony size are the longer and slender types encountered in the SEMS and expression of juvenile colony stage and fast growth? It is important to distinguish when M. erythraea or M. biserialis is meant, in several occasions (e.g. Line 276) both are combined as “Red Sea specimens”.

For the stable isotope discussion, it would be beneficial to dwell on some basic questions first, like: When exactly are sclerites formed? Is there a specific season of deposition? How fast are they formed? Do they continue to grow for extended periods after initial deposition? Can they be multi-annual? These questions are important because they relate to growth rates and vital effects, which affect the isotopic composition, while on the contrary the necessary bulk sampling of multiple sclerites to generate enough sample powder would iron out some of these effects. The aquaria culturing work by Taubner et al. (2012) would be a good starting point to address these questions and implications for the current work.
At the current state the oxygen isotopes are directly used to calculate temperatures at the time of deposition and since the values fall into the measured temperature range they are accepted as correct. Most octocorals have massive vital effects both in there calcareous skeletons and in the sclerites that can confound this interpretation. It would be better to start from the expected calcite d18O at equilibrium based on water temperature and seawater d18O composition, for each sclerite batch and measured temperature, and rigorously see if they coincide or by how much they are offset. Figure 5 is the most interesting in this aspect since both FWHM and d13C are correlated to sclerite axis length. FWHM is however also an expression of Mg-content – bring these data from the online materials to the results and introduce them in the discussion. These data have the potential to evaluate the degree of the present metabolic and kinetic vital effect. The Mediterranean C. rubrum would be indeed a suitable model organism for a comparison, as it has the same high-Mg calcite mineralogy and potentially similar Mg/Ca to d13C systematics (Chaabane et al. 2016, 2019; Le Goff et al. 2017; Taubner et al. 2012; Vielzeuf et al., 2018; Weinbauer et al., 2000). In Corallium rubrum juvenile fast grown skeletal axis (medullar zone and initial parts of annual growth rings) tend to have higher Mg-content and depleted d13C and d18O (Chaabane et al. 2016, 2019). Vielzeuf et al. (2018; see their Fig. 9) demonstrated further a systematic geochemical offset between skeleton and sclerites for Mg-content. It is hence conceivable that similar vital effects act also in the sclerites of M. erythraea. For comparison with Corallium rubrum sclerite isotopes see also Le Goff et al. (2017), and Taubner et al. (2012) for cultured alcyonarian sclerites. Weinbauer et al. (2000; Mar. Biol 137, 801-809) attempted a Mg-Temperature calibration in C. rubrum, this attempt has been used with mixed-success by Matsumoto (2007) in Primnoa pacifica. However, they observed that the sclerites collected at 0.65°C had similar compositions than the Mg-content in skeletons of Mediterranean C. rubrum collected at ~15°C, potentially impeding a temperature calculation. With your data and the systematic offset (skeleton to sclerites) observed by Vielzeuf et al. (2018), can actually provide an important mosaic stone to address these questions.
Overall, your data and direct interpretations with reconstructed temperature and degree of metabolic effects require a more rigorous discussion.

Suggested additions to references:
Chaabane, S., López Correa, M., Ziveri, P., Trotter, J., Kallel, N., Douville, E., McCulloch, M., Taviani, M., Linares, C., Montagna, P. (2019): Elemental systematics of the calcitic skeleton of Corallium rubrum and implications for the Mg/Ca temperature proxy.- Chemical Geology 524, 237-258. doi:10.1016/j.chemgeo.2019.06.008

Chaabane, S., López Correa, M., Montagna, P., Kallel, N., Taviani, M., Linares, C., Ziveri, P. (2016): Exploring the oxygen and carbon isotopic composition of the Mediterranean red coral (Corallium rubrum) for seawater temperature reconstruction.- Marine Chemistry 186, 11-23; doi:10.1016/j.marchem.2016.07.001.

Le Goff, C., Tambutté, E., Venn, A.A., Techer, N., Allemand, D., Tambutté, S. (2017): In vivo pH measurement at the site of calcification in an octocoral.- Scientific Reports 7:11210; doi:10.1038/s41598-017-10348-4.

Taubner, I., Böhm, F., Eisenhauer, A., Garbe-Schönberg, D., Erez, J. (2012): Uptake of alkaline earth metals in Alcyonarian spicules (Octocorallia).- Geochimica et Cosmochimica Acta 84, 239-255.

Titschack, J., Goetz-Neunhoeffer, F., Neubauer, J. (2011): Magnesium quantification in calcites [(Ca,Mg)CO3[ by Rietveld-based XRD analysis: revisiting a well-established method.- American Mineralogist 96, 1028-1038.

Vielzeuf, D., Gagnon, A.C., Ricolleau, A., Devidal, J.-L., Balme-Heuze, C., Yahiaou, N., Fonquernie, C., Perrin, J., Garrabou, J., Montel, J.-M., Floquet, N. (2018): Growth kinetics and distribution of trace elements in precious corals.- Frontiers in Earth Science 6:167; doi:10.3389/feart.2018.00167.

Weinbauer, M.G., Brandstätter, F. & Velimirov, B. (2000): On the potential use of magnesium and strontium concentration as ecological indicators in the calcite skeleton of the red coral (Corallium rubrum).- Marine Biology 137, 801-809; doi:10.1007/s002270000432.

Detailed comments
Figures
Figure 1: Inset Map – You might want to zoom to show the Eastern Mediterranean and the Red Sea, marking the distribution of Melithaea erythraea? Please add a km-scale.
Figure 1: Caption – mention where M. erythraea and where M. biserialis has been sampled. Mention SST-data source for Tel Aviv “Hadera” and Aqaba “Eilat”.
Figure 2: Are the Eilat sclerites in (A) from another M. biserialis specimen or from a thick morphotype of M. erythraea? Not immediately clear that the latter is the case - use letters (A) to (E) in caption.
Figure 2: Caption – change “sclerites morphologies” to “sclerite morphology”. Make caption more informative, pointing out thicker and shorter Red Sea sclerites with respect to slender Mediterranean forms. Point out also differences in wart-density.
Figure 3: In (A) only M. erythraea data are shown. M. biserialis is mentioned in the caption, but not plotted, this is not immediately clear. I presume the M. biserialis data have been omitted in (A) due to complete overlap with the M. erythraea data from Nasholim and Hadera? It would however be good to add them in (B) and (C).
Figure 4: change “EMS” with “SEMS”. It will be better to show raw sclerite δ18Osc values on the x axis, maybe with seawater δ18Osw correction – but putting “temperature of calcification” is an over interpretation of the data, since vital effects are rather common in octocoral calcite. An expected equilibrium range for high-Mg calcites for SEMS and Red Sea areas could be given instead?
Figure 5: (B) change y-axis label “δ13C” to superscript “δ13C”.

Raw Data Tables
Please introduce units in each column header.

Title/Abstract
Line 15: change “Correspondent” to “Corresponding”.
Line 31: change “of” to “for”.
Line 34: change “indigenous” with “endemic”.
Line 35: change “sclerites” to “sclerite”.

Introduction
Line 59: change “Hewitt et al., 2002” to “Hewitt and Hayes, 2002”.
Line 66: change “Bethoux” with “Béthoux”.
Line 69: change “Chevaldonne” with “Chevaldonné”.
Line 71: “tropicalization” lower case.
Line 73: check year “Peleg et al. 2020” – given as “2019” in references.
Line 77: change “Sisma-Ventura Yam” to “Sisma-Ventura, Yam”.
Line 86: change “zooxanthellae” with “zooxanthellate corals”.
Line 90: change “(Grossowicz et al., accepted)” to “(Grossowicz et al., 2020)”.
Line 91: change “might contributed” to “might have contributed”.
Line 94ff: you might want to support sentence with reference on sclerites.
Line 95: change “octocoral” to “octocorals”.
Lines 94 to 96: sentence needs a supporting reference (e.g. Taubner et al., 2012, GCA)
Line 96: change “Sclerites features” to “Sclerite features”. (check this plural issue throughout)
Line 100: change “M. erythraea’s” to “M. erythraea” (Italics).
Line 101: change “octocoral’s physiological traits” to “physiological traits of each octocoral”
Line 111: remove “soft”, distal axis is organic, but the basal stem turns into a massive calcitic stem.
Line 111: change “where magnesium” to “where the magnesium”

Methods
Line 121: change “sclerites” to “sclerite”.
Line 129: rephrase sentence. Mention species of 2002 specimen and collection site (not clear if Eilat or SEMS).
Line 134: change “debris were removed” to “debris was removed”.
Line 152: what is IRMS? Spell abbreviations out at first mention, check throughout text.
Lines 160/161: rephrase “at the stable isotope laboratory of the”.
Lines 160/161: change “Max-Planck institute” to “Max-Planck Institute”.
Line 163: change “70 degrees Celcius” to “70°C”.
Line 194: change “Sisma-Ventura, Yam and Shemesh, 2014” to “Sisma-Ventura et al., 2014”.
Line 200: FWHM – spell abbreviations out at first mention.
Line 200: These are geometric angles, remove Celsius from the unit.
Line 206: change “Site” to “site”.

Results
Line 215: change “sclerites” to “sclerite”.
Line 216: change “sclerites axis” to “sclerite axis”
Line 222: change “comparing” to “compared”.
Line 252: change “ranges” to “ranging”.

Discussion
Line 275: lower case “sclerite”.
Line 307: change “Maier, Paetzold,...” to “Maier et al.”
Line 309: “samples of 2016”, please state if they are from Hadera and/or Nasholim.
Line 316: change “Maier, Paetzold,...” to “Maier et al.”.
Line 323: change “(Grossowicz et al., accepted)” to “(Grossowicz et al., 2020)”.
Line 330: Aspect of crystal size needs to be explained in Material and methods already.
Line 331: “higher density” – density of warts on sclerite surface?

Conclusions
Line 339: change “Sisma-Ventura, Yam and shemesh, 2014” to “Sisma-Ventura et al., 2014”.
Line 339/340: change “Amitai et al., in review” to “Amitai et al., 2020”.

References
Line 361ff: Update reference to: “Amitai, Y., Yam, R., Montagna, P., Devoti, S., López Correa, M., Shemesh, A. (2020): Spatial and temporal variability in Mediterranean climate over the last millennium from vermetid isotope records and CMIP5/PMIP3 models.- Global and Planetary Change 189, Article #103159; doi:10.1016/j.gloplacha.2020.103159.”
Line 366: replace “(2007.)” by “(2007).”
Line 383: replace “Chevaldonne” with “Chevaldonné”.
Line 385: replace “D.” with “D.S.”
Line 386: replace “Methids Researc” with “Methods Research”
Line 399: update reference to “Grossowicz M, Shemesh E, Martinez S, Benayahu Y, Tchernov D (2020). New evidence of Melithaea erythraea colonization in the Mediterranean. Estuarine, Coastal and Shelf Science 236; doi:10.1016/j.ecss.2020.106652.”
Line 403: replace “Chemical Geological” with “Chemical Geology”, remove “Isotope Geoscience”
Line 410: add page numbers, or “doi:0.3389/fmars.2018.00001”.
Line 412: replace “europe” with “Europe”.
Line 432: remove “676”.
Line 448: replace “Paetzold” with “Pätzold”.
Line 452: replace “Porites Lutea” with “Porites lutea”.
Line 458: check year “Peleg et al. 2019” – as “2020” in text.
Line 459: replace “ecology” with “Ecology”.
Line 464f: add reference for “R Core Team, 2014” mentioned on line 212.
Line 465: remove “et al.” add authors “Teocharis A and Papathanassiou E”.
Line 478: check page numbers
Line 503: Italics “Acabaria biserialis”.

Review by M. López Correa (CNR-ISMAR, Bologna, Italy)

---

## Round 0.2 · accepted · Accept

Dear Michal and co-authors,

I am delighted to accept your manuscript for publication in PeerJ. I have added a few comments in the 'editor_annotated' pdf for incorporation at the proofs stage, as you see fit - all very minor.

Thank you for this great scientific contribution, and looking forward seeing it published!

With warm regards,
Xavier

Reviewer 1 ·

Basic reporting

The manuscript is now ready to be published

Experimental design

The manuscript is now ready to be published

Validity of the findings

The manuscript is now ready to be published

Additional comments

The manuscript is now ready to be published

·

Basic reporting

The work by Grossowicz et al. on the octocoral Melithaea erythraea has carefully incorporated the comments by several reviewers, seen language polishing, and has significantly ameliorated and can now be recommended for publication with PeerJ in its current form.

Experimental design

Review comments have been accomodated.

Validity of the findings

Review comments have been accomodated.